# The unequal burden of human-wildlife conflict

Alexander R. Braczkowski[1,2,3], Christopher J. O'Bryan [4,5], Christian Lessmann[6,7], Carlo Rondinini[8],
Anna P. Crysell[9], Sophie Gilbert[10,11], Martin Stringer[12], Luke Gibson [1✉] & Duan Biggs[2,13,14]

Human-wildlife conflict is one of the most pressing sustainable development challenges globally. This is particularly the case where ecologically and economically important wildlife impact the livelihoods of humans. Large carnivores are one such group and their co-occurrence with low-income rural communities often results in real or perceived livestock losses that place increased costs on already impoverished households. Here we show the disparities associated with the vulnerability to conflict arising from large carnivores on cattle (*Bos taurus*) globally. Across the distribution of 18 large carnivores, we find that the economic vulnerability to predation losses (as measured by impacts to annual per capita income) is between two and eight times higher for households in transitioning and developing economies when compared to developed ones. This potential burden is exacerbated further in developing economies because cattle keepers in these areas produce on average 31% less cattle meat per animal than in developed economies. In the lowest-income areas, our estimates suggest that the loss of a single cow or bull equates to nearly a year and a half of lost calories consumed by a child. Finally, our results show that 82% of carnivore range falls outside protected areas, and five threatened carnivores have over one third of their range located in the most economically sensitive conflict areas. This unequal burden of human-carnivore conflict sheds light on the importance of grappling with multiple and conflicting sustainable development goals: protecting life on land and eliminating poverty and hunger.

[1] School of Environmental Science and Engineering, Southern University of Science and Technology, Shenzhen, China. [2] Resilient Conservation, Centre for Planetary Health and Food Security, Griffith University, 170 Kessels Rd, Nathan, QLD 4111, Australia. [3] School of Natural Resource Management, Nelson Mandela University, George Campus, Madiba Drive, 6530 George, South Africa. [4] School of Earth and Environmental Sciences, The University of Queensland, St Lucia, QLD 4067, Australia. [5] Centre for Biodiversity and Conservation Science, University of Queensland, St Lucia, QLD 4067, Australia. [6] Technische Universität Dresden, 01069 Dresden, Germany. [7] Ifo Institute & CESifo, Poschingerstr. 5, 81679 Munich, Germany. [8] Center for Global Wildlife Conservation, State University of New York College of Environmental Science and Forestry, Syracuse, NY 13210, USA. [9] Department of Political Science, University of California Los Angeles, Bunche Hall, 4289 Los Angeles, USA. [10] Nature Capital Development, 443 Fillmore Street 380-1418, San Francisco, CA 94115, USA. [11] Affiliate faculty, Department of Fish and Wildlife Sciences, University of Idaho, Moscow, ID 83843, USA. [12] W.H. Bryan Mining and Geology Research Centre Sustainable Minerals Institute, The University of Queensland, Level 4, Sir James Foots Building, St Lucia, QLD 4067, Australia. [13] Olajos-Goslow Chair of Environmental Science and Policy, Northern Arizona University, 624 Knoles Dr, Flagstaff, AZ 86011, USA. [14] Centre for Complex Systems in Transition, School of Public Leadership, Stellenbosch University, 19 Jonkershoek Rd, Mostertsdrift, Stellenbosch 7600, South Africa. ✉email: biodiversity@sustech.edu.cn

One of Earth's foremost challenges to sustainable development is ensuring that the human communities living alongside wildlife and protected areas prosper[1–3]. While wildlife such as large carnivores—species ranging from wolves and hyenas to lions and bears—are critical for regulating ecosystem processes that improve human well-being[4–7] and contribute substantially to tourism economies[8,9], their protection is challenging because they often kill valuable livestock in both community lands and areas adjacent to protected areas[10–13]. The economic shocks of losing livestock to large carnivores can be very high[14], because as much as two-thirds of a household's annual income can be lost in a single livestock predation event[15]. For example, families living inside Jigme Sigmye National Park in central Bhutan lost on average 17% of their yearly per-capita income from tiger (*Panthera tigris*) and leopard (*Panthera pardus*) predation[16], and those on the edge of Tanzania's Serengeti National Park lost over 19% annually from leopard and lion (*Panthera leo*) predation[17]. Moreover, some 750 million to one billion livestock keepers own limited livestock, are landless, live on less than two US dollars per day[18,19], and have pastureland overlapping some of the least productive vegetation zones[20]. In lower latitudes, livestock are also kept as a risk management asset[21]. The financial shocks from human–wildlife conflict often disproportionately affect such producers[15] and are an additional source of stress in communities already impacted by climate change, armed conflict, and disease events[22–24].

In some areas carnivores are valued for cultural and religious reasons[25–27] but where these are absent conflict between agricultural communities and carnivores often escalates carnivore extinction risk due to the retaliatory killings following livestock predation[28–31]. For example, intense conflicts between cattle farmers and African lions in WAZA National Park in Cameroon and Queen Elizabeth National Park in Uganda have led to unsustainable rates of retaliatory killings over the past three decades, causing severe declines in these lion populations[32,33]. Such retaliatory killings are important for a series of reasons including that they negate the increasingly important role carnivores play in both trophic regulation[34,35], and the direct benefits they provide to humans[5]. For example, high mule deer (*Odocoileus hemionus*) densities and reduced tree recruitment in the Zion and Yosemite National Parks USA were linked to localized extinctions of mountain lions (*Puma concolor*)[36,37]. Similarly, mountain lions and gray wolves (*Canis lupus*) make roadways for humans safer in South Dakota and Wisconsin USA through reducing vehicle collisions with deer and save millions of dollars in resulting insurance costs and hospital fees[7,38]. Given that rangelands make up over half of the earth's terrestrial surface[39] and that agricultural lands are projected to increase in extent by 2–10 million km$^2$ (~15–20%) in the coming decades[40], large carnivores are at unprecedented risk of extinction along with the unique ecosystem services they provide.

While research on the negative economic impacts of large carnivore predation on livestock has been ongoing at local scales[41,42] including some well-developed examples[43,44], and even those examining spatial human-wildlife conflict risk at the continental scale[45], no work to date has comparatively explored the potential economic burden of large carnivores globally. Here we present the first spatially explicit analysis of the potential economic burden (Supplementary Box 1) arising from human-wildlife conflict at the global scale illustrating the financial and social costs of losing livestock on human communities. We do this by examining the proportion of GDP per capita (hereafter referred to as per capita income) that is vulnerable from the loss of a single cattle calf (the equivalent of 250 kg or one tropical livestock unit[46]) in 133 countries that overlap with the habitat of 18 large carnivore species known to prey on cattle (adjusted for the presence of cattle). We then assess the direct and opportunity costs of lost calories to households from the predation of a single cow or bull and contrast this against the production yield of meat per animal. Finally, we examine the proportion of carnivore range found within highly economically sensitive conflict areas (i.e., areas where communities would experience ≥25% loss to per capita income during a predation event, we define these areas as frontline communities) and discuss this in the context of a suite of already highly threatened carnivore species. Our study shows the predicament facing economically fragile households and carnivore species occupying the same habitats, thereby illustrating the difficult tradeoffs between three closely interlinked United Nations Sustainable Development Goals: no poverty (Goal 1), zero hunger (Goal 2), and protecting life on land (Goal 15).

## Results

**Potential burden hotspots**. Using an analysis based on subnational 1st level administrative regions our data show that the world's poorest people may bear the highest cost of living with large carnivores (Fig. 1). This is based on the disparities in economic vulnerability to carnivore predation on cattle. People living in developing countries will on average experience an eightfold higher potential economic burden ($\bar{x}$ = 32%, range = 0.02–201% of per capita income lost in a large carnivore predation event) than those living in developed economies ($\bar{x}$ = 4% income lost, range = 1–9%, Supplementary Table 2). Moreover, 13% of the countries ($n$ = 17 out of 133) in carnivore range globally are potentially under threat of losing more than half of their per capita income. The most economically sensitive countries (i.e., those with the highest potential economic burden where >50% economic income loss would occur through predation) in our global assessment were the African states of Mozambique, Guinea-Bissau, Malawi, Tanzania, Uganda, and Burundi, and the Southeast Asian countries of Cambodia and Laos. In these countries human communities are vulnerable to losing all or more than double their annual income ($\bar{x}$ = 132%, range = 106–201% of per capita income) if a single calf is killed by carnivores. The situation is different in some of the most economically developed countries like Sweden, Canada, United States, Australia, and Spain where the vulnerability of a single predation event is nominal ($\bar{x}$ = 1.67%, range = 1.29–1.92% of per capita income, Supplementary Table 2).

**Carnivore species in highly sensitive conflict areas**. The geographic distribution of the 18 large carnivores in our analysis was 72,313,995 km$^2$ in 2009, and 82% of this fell outside of protected areas. Nearly a quarter of all carnivore range in our analysis (23%) fell within the highest conflict burden areas (≥25% per capita income vulnerable to a single predation event). Ten species had more than a third of their range in areas where a predation event would represent a severe economic burden (Table 1), and this included eight species considered globally threatened by the IUCN. These species had a mean proportion of 61% of their range overlapping areas where a cattle predation would amount to ≥25% of per capita income lost. The species with the highest range overlap were the Snow leopard *Panthera uncia* (89% range overlap, Fig. 1), African lion *Panthera leo* (78% range overlap), Asiatic black bear *Ursus thibetanus* (70%), striped hyena *Hyaena hyaena* (66%), and leopard *Panthera pardus* (64%, Table 1). The species with the lowest range overlap with highly sensitive conflict areas were the dingo *Canis lupus dingo* (0% range overlap), red wolf *Canis rufus* (0%), American black bear *Ursus americanus* (0%), puma *Puma concolor* (1%), and jaguar *Panthera onca* (4%, Table 1).

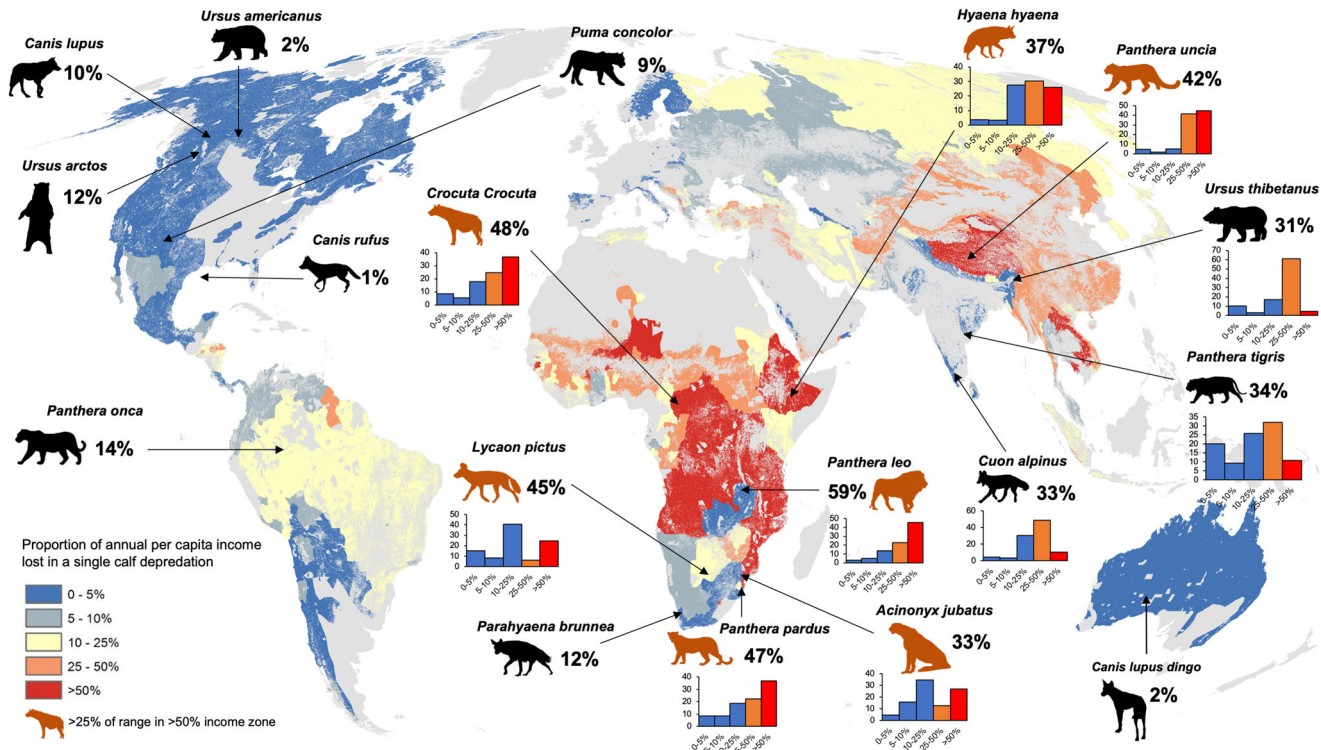

**Fig. 1 The average annual per capita income percentage loss recorded across the range of 18 large carnivores globally under a single calf predation event.** Our model assumes that a single cattle keeper is exposed to a single calf (250 kg) predation event anywhere across a given carnivore's range. The orange silhouettes represent those species that have >25% of their range located in areas where communities would experience >50% economic income loss through predation. Bar charts are provided for the ten species of large carnivore identified in our analysis that have more than a third of their range in areas where a conflict event would represent a high economic burden (i.e., ≥25% of per capita income vulnerable from a single predation event). Silhouettes obtained from www.phylopic.org and are used under the Creative Commons Attribution-Non Commercial 3.0 Unported license. Silhouette credits are as follows: *Puma concolor* = Cristian Osorio & Paula Carrera, *Ursus americanus, Ursus thibetanus* and *Ursus arctos* = Tracy Heath, *Canis lupus, Panthera pardus, Acinonyx jubatus, Hyaena* hyaena, *Parahyaena* brunnea, and *Panthera leo* = Margot Michaud, *Panthera* onca, *Panthera uncia* and *Lycaon pictus* = Gabriela Pamono-Munoz, *Canis lupus dingo* = Sam Fraser-Smith, *Panthera tigris* = Steven Traver, *Cuon alpinus* = Michael Keesey, *Canis rufus* = David Orr, *Crocuta crocuta* = Oscar sanisidro.

**Food security impacts**. Predation events also represent a loss in direct and opportunity costs to cattle keepers. Assuming a single predation of a cow or bull, this can result in 227.33–1229.07 immediate usable kilocalories (1kcal = 1000 calories) lost to a cattle keeper and their family (Fig. 2). This caloric loss is highest in the developed world ($\bar{x}$ = 817.80 kcal, range = 478.17–1130.22) but most severe in developing economies because daily per capita meat intake is 37% lower than in the developed world (Supplementary Table 4). The immediate caloric losses from a single predation event in developing economies translates to nearly 1.5 years ($\bar{x}$ = 1.42, range = 0.77–3.12) of lost daily calories required by a 2–3-year-old child, and roughly two thirds of a year ($\bar{x}$ = 0.69, range = 0.26–1.41) for a 12–13-year-old adolescent, and 30–59-year-old adult human ($\bar{x}$ = 0.66, range=0.25–1.35). We estimate another 131.04 and 33.88 kcal are lost annually from potential milk and meat production respectively under the assumption that 312 kg of milk and 12.5 kg of meat are produced annually[47].

**Economic disparities in national cattle production**. We found that cattle prices globally were comparable for the 2009 year ($F_{(2,76)}$ = 0.41, $p$ = 0.66, Fig. 3), the same year as our AOH habitat carnivore data. There was also no significant difference in cattle price between developing, transition, and developed economies for multiple years beginning in 1961 ($F_{(2,103)}$ = 0.32, $p$ = 0.73, Fig. 3). However, we observed disparities in the meat yield per cattle between these different economic groups with developing

(174.05 kg/animal, SD = 63.39) and transition (153.4 kg/animal, SD = 36.99) economies producing significantly less meat per animal ($F_{(2,131)}$ = 20.75, $p$ = <0.00001) when compared to developed states (253.98 kilograms/animal, SD = 54.86) for the 2009 year, and over multiple years with developing (155.91 kilograms/animal, SD = 53.97) and transition (149.87 kg/animal, SD = 37.11) economies producing significantly less meat per animal ($F_{(2,6983)}$ = 827.088, $p$ = 0.00) when compared to developed states (223.94 kg/animal, SD = 57.88).

**Discussion**

Our results highlight two key findings, namely that Earth's poorest communities pay the highest financial and food security costs for conflict (it is eight orders of magnitude worse for a cattle keeper, as a proportion of their per capita income, to lose a calf from conflict in a developing country than a developed one), and more than half of the world's largest carnivore species have over one third of their habitat overlapping the most economically vulnerable human communities (what we term economic front-line communities). Our results confirm that not only is this true in absolute per capita income loss, but also due to cattle keepers in developing economies producing less meat per animal and having a lower meat intake per capita. This is notable because in many areas of the developing world cattle keepers are already under immense pressure from localized rainfall patterns, drought, and climate change[21,48], and the fragmentation of once contiguous pasture lands[49]. This is causing livestock units per capita

**Table 1 The proportion of Area of Habitat (AOH) of 18 large carnivore species within low, intermediate, high, and severe vulnerability related to a calf depredation event. This area represents the percentage of each species range that falls within the most economically sensitive communities on earth.**

| Scientific name | IUCN status | Mean annual income loss (%) | % Range in low-vulnerability area | % Range in intermediate vulnerability area | % Range in high-vulnerability area | % Range in severe vulnerability area |
|---|---|---|---|---|---|---|
| *Panthera uncia* | VU | 41.68 | 4.59 | 1.75 | 5.13 | 88.52 |
| *Panthera leo* | VU | 59.07 | 2.78 | 5.16 | 13.76 | 78.29 |
| *Ursus thibetanus* | VU | 31.26 | 10.09 | 2.94 | 17.18 | 69.79 |
| *Crocuta crocuta* | LC | 47.57 | 8.38 | 5.40 | 18.07 | 68.15 |
| *Hyaena hyaena* | NT | 36.64 | 3.67 | 3.17 | 27.60 | 65.57 |
| *Panthera pardus* | VU | 46.81 | 8.59 | 8.50 | 18.56 | 64.35 |
| *Cuon alpinus* | EN | 33.01 | 4.18 | 3.35 | 30.53 | 61.93 |
| *Acinonyx jubatus* | VU | 32.51 | 4.63 | 15.79 | 34.54 | 45.04 |
| *Panthera tigris* | EN | 33.57 | 20.09 | 9.34 | 25.74 | 44.83 |
| *Lycaon pictus* | EN | 44.79 | 15.32 | 8.28 | 40.55 | 35.86 |
| *Ursus arctos* | LC | 11.63 | 24.70 | 18.66 | 48.22 | 8.42 |
| *Canis lupus* | LC | 10.02 | 33.48 | 17.92 | 40.73 | 7.86 |
| *Parahyaena brunnea* | NT | 12.27 | 18.93 | 55.13 | 22.03 | 3.91 |
| *Panthera onca* | NT | 13.50 | 12.18 | 15.70 | 68.57 | 3.55 |
| *Puma concolor* | LC | 8.60 | 44.33 | 14.37 | 39.89 | 1.41 |
| *Canis lupus dingo* | LC | 1.90 | 100 | 0.00 | 0.00 | 0.00 |
| *Canis rufus* | CE | 100 | 0.00 | 0.00 | 0.00 | 0.00 |
| *Ursus americanus* | LC | 1.54 | 99.14 | 0.86 | 0.00 | 0.00 |

Classifications are as follows – low vulnerability = 0–5% per capita income loss, intermediate vulnerability = 5–10% per capita income loss, high vulnerability = 10–25% per capita income loss, and severe vulnerability = >25% per capita income loss.

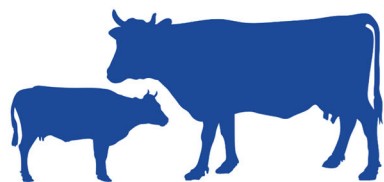

### A. Direct Costs of Losing a Single Cow or Bull

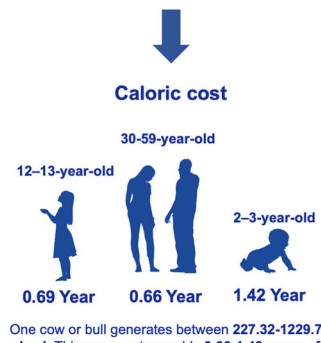

The costs of losing a single cow or bull to a household in a developing economy can be catastrophic. **Apart from economic costs, there are immediate caloric costs that are incurred by households.**

**Caloric cost**

30-59-year-old

12–13-year-old

2–3-year-old

0.69 Year 0.66 Year 1.42 Year

One cow or bull generates between **227.32-1229.70 kcal**. This represents roughly **0.66-1.42 years of human caloric requirements**[109] in developing economies.

### B. Opportunity Costs of Losing a Single Cow or Bull

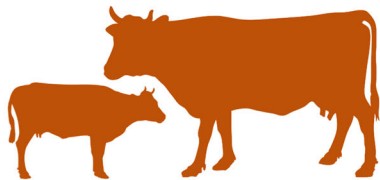

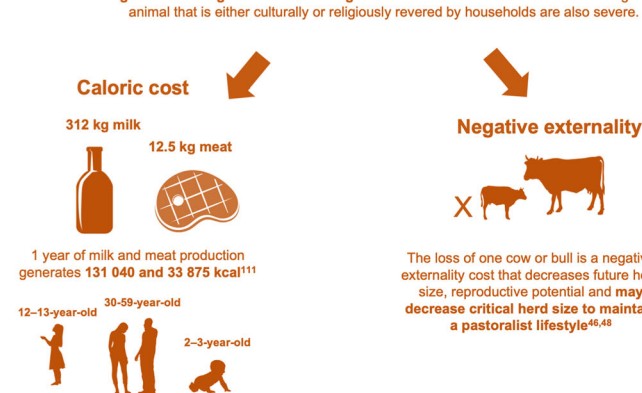

The costs of losing a single cow or bull transcend direct financial and caloric costs. There are significant opportunity costs. **Over a 1-year period we estimate that a cow could generate 312 kg of milk and 12.5 kg of meat offtake**[47]. Costs attributed to losing an animal that is either culturally or religiously revered by households are also severe.

**Caloric cost**

312 kg milk 12.5 kg meat

1 year of milk and meat production generates **131 040 and 33 875 kcal**[111]

12–13-year-old 30-59-year-old 2–3-year-old

1.8 Months 1.7 Months 4 Months

**Negative externality**

X

The loss of one cow or bull is a negative externality cost that decreases future herd size, reproductive potential and **may decrease critical herd size to maintain a pastoralist lifestyle**[46,48]

**Fig. 2 The direct and opportunity costs of losing a single cow or bull in areas of low per capita income. A** The direct costs of losing a single cow or bull, and B the opportunity costs of losing a single cow or bull. Costs are calculated in direct caloric. Data and references are taken from 47 and 109. Lost opportunity cost may also be manifested as a negative externality whereby a lost cow or bull decreases herd size, reproductive potential, and places livestock producers where they can no longer maintain a pastoral lifestyle as noted in 46 and 48. Silhouettes obtained from www.phylopic.org and www.freepik.com. Silhouette credits are as follows: bottle and meat = macro_vector on Freepik, human silhouttes = www.publicdomainvectors.com, cattle = Steven Traver and Andreas Preuss.

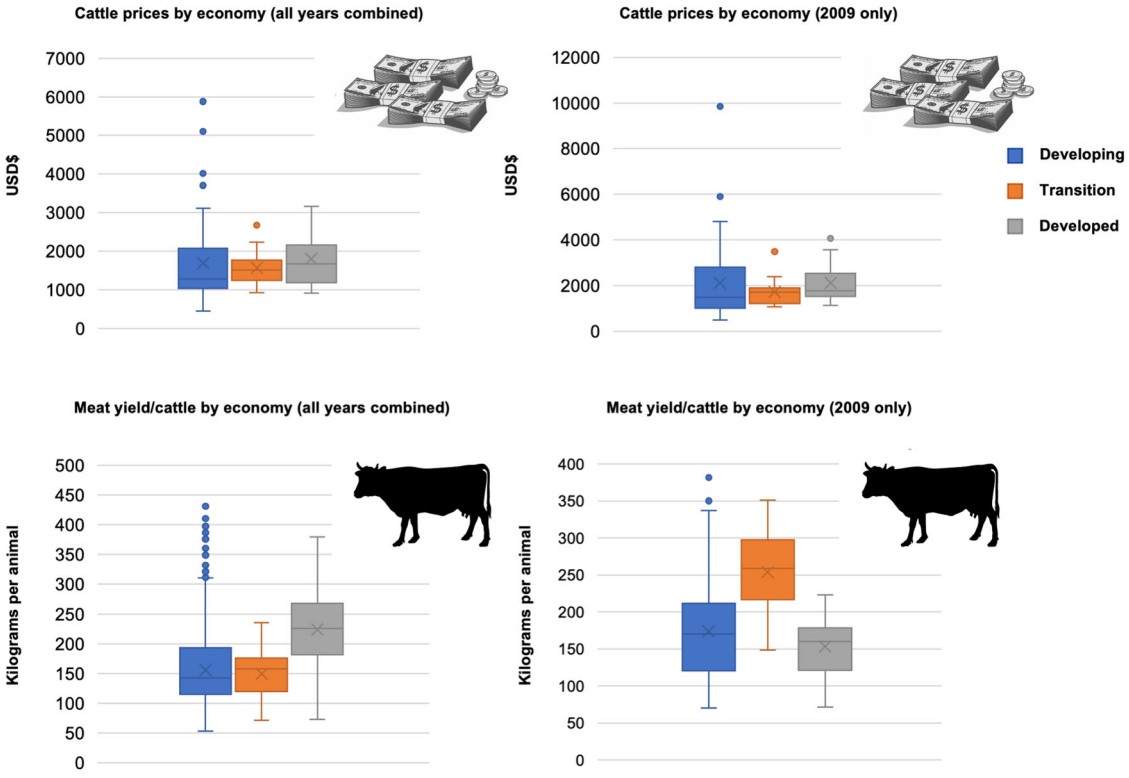

**Fig. 3 Box plots representing variation in cattle price variation and cattle meat yield per animal according to either developing, transition, or developed economies.** We provide data for both the 2009 year (which corresponds to our AOH carnivore habitat data), and for all years since 1961. Silhouettes obtained in public domain.

to decline[50]. This in turn has a knock-on effect on herd size that when below a critical threshold leads pastoralism to no longer be viable[46,48]. Estimates vary but typically if a cattle keeper owns two or fewer adult cattle (4.5 tropical livestock units equivalent to 1125 kg of livestock biomass per capita) they are unable to maintain a nomadic cattle keeping lifestyle[48]. Any source of livestock loss at this point is too severe for economic recovery[46] and will potentially erase the wealth of an entire cattle keeping family. These income losses may have a knock-on effect on food security, and this may disproportionately affect women who tend to forgo meals to keep the household fed[51,52]. This mirrors crop raiding scenarios like those highlighted in the Kavango-Zambezi Transfrontier Conservation Area[53]. These authors showed that elephant-associated crop raiding exacerbated the likelihood of food insecurity due to an already reduced rainfall environment. We anticipate that such knock-on effects we discuss here are likely experienced by many if not most of the households that would lose ≥25% of per capita income in our analysis. Moreover, further knock-on effects are likely, such as migration, subsequent social, and political instability, and impacts to childhood cognitive skills and education due to loss of calories[54]. Our analysis also illustrates two sources of additional and lost opportunity cost, that of useable calories from meat, and those garnered from milk and associated products. We estimate the immediate useable calories to be somewhere between roughly half a year and 1.5 years of calories depending on human age, while we estimate an additional ~300 kg of milk and 12 kg of meat/year from cattle yield. These kinds of costs ignore additional valuations of other cattle products such as dung used for fertilizer[55,56], the hidden mental costs associated with the loss of a sacred animal in some regions[57], and critical thresholds of cattle herd size[48].

Charismatic carnivore species like African lions and tigers are highly valuable to societies, and this is especially true at the

national or global level[58,59], but at more local scales they often have little or even negative value[9,15]. Despite the lamentations of western governments and celebrities over declining carnivore populations, the global community suffers little to no personal cost from large carnivores. This lies in stark contrast to the experiences of local communities living alongside these predators in the developing world for whom a single predation event represents on average a third of their annual income. It is these local communities who "are burdened disproportionately with global wildlife maintenance despite the commonplace notion of wildlife resources as the birthright of people everywhere"[60] (pp. 35–36). Furthermore, outside actors from developed nations at times impede conservation efforts on reserve edges and community lands in developing nations, such as in the case of trophy hunting bans that deny communities a proven mechanism to benefit economically from wildlife[61,62].

Our results arrive at a critical juncture, where calls for environmental conservation are increasingly tempered by recognition of the environmental justice concerns that go with such requests[60,63,64]. The poorest 40% of humanity suffer impacts from climate change 70% greater than the global average[65,66]. Likewise, our analysis shows that human communities lying outside of protected areas overlap with 82% of the Earth's large carnivore range, whilst also bearing potentially the heaviest economic burdens of living alongside them. Building on work highlighting the tradeoffs between biodiversity conservation and food production[63], we highlight the at times conflicting nature of three of the United Nations Sustainable Development Goals (SDGs): no poverty (Goal 1), zero hunger (Goal 2), and protecting life on land (Goal 15). In the context of human-wildlife conflict these three SDGs are often in direct competition with one another, mainly because there are no sufficient bridging mechanisms for communities to access the potential value of

damage-causing but charismatic species. In cases where human communities living alongside protected areas and damage-causing wildlife realize the monetary value from these species (e.g., through ecotourism revenues, conservancies, and population performance payments), and this revenue is not subjected to elite capture or is marginal in nature[67,68], pressures on protected areas and wildlife wane. This is due to increases in human development indices and people escaping from poverty traps[69–71]. Indeed, human-wildlife conflict has also been shown to reduce when non-economic conflict reduction models are used. These include *inter alia* the introduction of human and canine guardians for livestock[41], and the building of protective enclosures and bomas to shield livestock against predators[72].

Our results illustrate the urgent need to create economic bridging measures that will reconcile the high value placed on carnivores globally or nationally, with the economic inequalities suffered by local communities. Recent developments in the novel conservation financing sector are an example of such mechanisms. Two examples include South Africa's "Rhino Bond" and Sweden's payment for wolverine presence. Both are tied to the population performance of these species in community lands and protected areas[73,74]. Conservancy models which promote tourism and revenue generation in non-protected areas but are not necessarily limited to non-consumptive-tourism only (still having a livestock component[75]) are another financial model for the problem we have identified here[76]. These solutions deal explicitly with the problem of damage-causing carnivore species (seven of which that are considered globally threatened by the IUCN Red list of threatened species) ranging on non-protected community land which is not only suitable habitat but is also co-inhabited by cattle. It should also be noted that the costs associated with predation events often pale when compared to climatic or disease-related shocks which run into the billions of US$ dollars[77].

Our analysis of the potential economic burden arising from losing livestock to carnivores was only possible due to recent developments in global socioeconomic[78] and carnivore range spatial data[79]. We feel however, that our analysis is extremely conservative for several reasons, namely we only show the potential economic ramifications of a single predation event. Carnivore depredation often manifests itself in specific areas due to habitat or ecological variables[80,81], households often experience predation events multiple times per year[15], and sometimes carnivores engage in surplus killing[82–84]. Our analysis does not calculate actual depredation rates, it only illustrates highly vulnerable (and buffered) economic areas globally that would suffer under a predation event. Our valuation of a predated calf is also likely low, because we adjusted the measure of FAO market cattle value to the slaughter weight of a single ~6-month-old calf (250 kg), equating to roughly 33% of the economic value of an adult cow, and our measure of a single calf as a proportion of per capita GDP is well below the threshold of total income generated from many cattle production systems in Africa and Asia (for instance across much of sub-Saharan Africa livestock typically contributes between half and all of household income in rural settings, with cattle contributing 70–90%; 47). We acknowledge that a lack of regional cattle price data means that our analysis of cattle market price is relatively coarse. Prices in livestock also change with the breed of cattle, seasonality (and drought) and with prevailing macro and micro economic conditions[85]. Further, our data originate from 2009 and the acceleration of human-induced climate change, the COVID-19 pandemic, and increased rates of political strife may have impacted the severity of our analysis even further.

How cattle keepers view and utilize cattle also varies sharply across the globe, and as[86] note in their key paper "there is an enormous variability in herd management strategies, in social organization, in land tenure, degree of dependence on agricultural products, interactions with outside groups, differentiations of sex and age, etc". We agree that such variation may be present even at incredibly fine spatial scales. For example, the Bahima and Karamajong cattle keepers of Uganda's Mbarara and Karamoja districts, place immense cultural and monetary value into cattle keeping. It is a central part of their identity[87]. They largely subsist off cow milk and blood, use cows as dowry or enzhugano[88], but rarely slaughter the cows for meat[89]. Contrastingly, South African Zulus slaughter a high quantity of cattle for consumption in their diet[90]. Similar contrasts can be observed in India. Roughly 80% of the human population is comprised of Hindus[91] which consider cattle as sacred and do not slaughter and consume cattle. This contrasts with the roughly 14% of Indians which are Muslim. Forty percent of these Muslims include cattle meat in their diet. Our analysis does not capture the nuances of these cultural or even religious valuations of cattle. We also do not differentiate the variations in production practice of cattle globally. For example, in the Brazilian Pantanal cattle ranches are large (over half are between 5 and 30,000 hectares in size[92], intensive in nature, and slaughter and export to the international market[93]. This contrasts with for example Maasai group ranches in southeast Kenya that own on average between 250 and 650 hectares depending on their location[94]. Our analysis encapsulates both highly productive intensive systems, and nomadic, subsistence production types.

Finally, our results only touch on the many potential opportunity costs (Fig. 3) stemming from conflict. There are a multitude of hidden costs[95] and perceived conflicts that can be associated with such a loss including increased workload to make up for financial losses[51], physical displacement of households[96], the physical and disease exposure risks of guarding livestock at night to prevent further losses[95], transaction costs of pursuing compensation payments, and the failure to obtain fair livestock value[97,98]. A recent example from Zimbabwe shows that the presence of an African lion equates to negative USD$180 per person per year due to fear, and a lack of trust in compensation by authorities responsible in mitigating conflict events[9]. There may also be a host of psychological effects stemming from such conflict including fear of attack by carnivores, hesitancy to move in the dark, grief over lost livestock, and even PTSD from livestock loss, not to mention the immense cost of human life itself[99].

## Methods

**Mapping large carnivore habitat**. We mapped the spatial habitat extent of 18 large carnivores known to prey on cattle (Supplementary Table 1) using Area of Habitat (AOH) data from[100], except for the dingo *Canis lupus dingo*, which was mapped following[101]. AOH represents areas of fine scale (300 m) high habitat suitability within large carnivore IUCN geographic distributions during the year 2009. AOH has been used in several recent global studies[102,103] due to its reduced risk for commission errors[79], fine spatial scale, and incorporation of heterogeneous environmental variables (e.g., land cover, elevation, and hydrological features[104]. These layers were used for subsequent analyses. Excluded countries are presented in Supplementary Table 3.

**Mapping per capita income**. We mapped annual per capita income at the sub-national scale (i.e., state or province scale) from for the year 2009[78]. We chose the year 2009 to correspond with the year of the large carnivore distribution data, as described above. We chose this dataset because these data provide high resolution estimates of per capita income for both developed and developing countries based on nighttime luminosity data[78]. Nighttime luminosity is important because household survey data (e.g., Afrobarometer or DHS) usually do not contain income data since households fear expropriation from the government and/or cannot provide a monetary equivalent of returns from agricultural production. Generally, as income rises, so too does electricity usage and subsequent nighttime light signature per person, in both production activities and consumptive ones[105], and light has been used as a proxy for income per capita in several previous studies[106–108].

**Estimating cattle prices**. We obtained data on national cattle prices per kilogram (Item Code 945, FAO) from the Food and Agriculture Organization of the United Nations[109]. This dataset reports on cattle prices as collected at the point of initial sale (prices paid at the farm-gate). Because our large carnivore AOH and per capita income data corresponded to the year 2009, we included the average cattle live weight price for the year 2009, or the next closest year available in the FAO database or gray literature (see Supplementary Data File 1 for details). We then determined the price of a single sub-adult cattle calf by multiplying the per kilogram cattle price for a given country by 250 kg, the approximate size of a sub-adult cattle calf, also known as a single tropical livestock unit[46]. This estimated price of a sub-adult cattle calf per country was used for estimating the financial burden of large carnivore predation, described below.

**Mapping cattle distribution**. We mapped the spatial distribution of cattle using the updated Gridded Livestock of the World (GLW 3[110]). The GLW 3 represents areas of sub-national livestock densities, including cattle at fine resolution (~10 km at the equator) for the year 2010. We used the dasymetric dataset of cattle, which corresponds to previous GLW datasets and represents different cattle densities per pixel within a census area according to random forest models. We determine cattle to be present in a given pixel if the density was greater than zero. We resampled the data (bilinear method) to match the spatial resolution of the large carnivore AOH data (300 m).

**Mapping burden hotspots**. We determined the financial burden of losing a single sub-adult cattle calf to large carnivore predation by first dividing estimates of the price of a single sub-adult cattle calf (see *Estimating cattle prices*) with estimates of per capita income at the sub-national scale (see *Estimating per capita income*). This produced the relative proportion of annual per capita income lost assuming the predation of a single sub-adult cattle calf. We then masked sub-national administrative boundaries, which contained the above information on the proportion of per capita income vulnerable, with the spatial extent of large carnivores (see *Mapping large carnivore habitat*)—that is, a spatial mosaic of all 18 large carnivore AOH. Next, we masked this layer with data[110] on the spatial distribution of cattle globally (see *Mapping cattle distribution*). Finally, we intersected the AOH (corrected with the distribution of cattle) with country boundaries to determine the per capita financial burden at the national scale, and we intersected this with individual carnivore AOH layers to determine the potential financial burden within each species' geographic range (Supplementary Data 3).

We report our results of potential economic burden for each country as an average annual per capita income loss across the entire country. We also calculated the proportion of each carnivore species' range overlapping areas that experience different levels of economic burden: 0–5% of per capita income loss (very low-vulnerability area), 5–10% (low vulnerability area), 10–25% (moderate vulnerability area), 25–50% (high vulnerability area), and >50% (extreme vulnerability area).

**Estimating food security impacts**. We also assessed the direct and opportunity costs lost to households from predation. We calculated calories lost from a single cow or bull (note the FAO data provides hectograms per animal for yield) predation through multiplying country-specific meat yield from the FAO 2021 by beef carcass kilojoule value (1351 kJ/100 g) (see: https://www.fao.org/ag/againfo/themes/en/meat/backgr_composition.html). We then divided these lost calories by the average estimated daily caloric intake of a young child aged 2–3, an adolescent aged 12–13, and an adult aged 30–60-years-old[111].

**Estimating economic disparities in national cattle production**. Finally, we assessed the disparities in cattle prices between developing, transition, and developed economies and examined the differences in meat yield per carcass (Supplementary Data 2). We did this because we wanted to ascertain whether cattle keepers in developing and transition economies may be further exposed (comparatively to developed states) to conflict due to a) price plasticity in cattle markets over time (these could be indicative of market or climate-related shocks[46,48]), and b) low productivity in meat production per animal (ie. prices of cattle per kg may be similar between countries however cattle keepers have to produce more cattle per unit area to yield the same price per ton). We used cattle meat yields in kilograms per animal obtained from the FAO agricultural database[109]. Potential disparities in a) the cattle price (both for the 2009 year, and over time using our historic FAO cattle price dataset), and b) meat yield per animal between developing, transitioning, and developed economies were assessed using a one-way analysis of variance ANOVA.

**Reporting summary**. Further information on research design is available in the Nature Portfolio Reporting Summary linked to this article.

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

## Author contributions
Conceptualization: A.P.C., A.R.B., C.J.O., C.L., D.B., L.G., S.G. Data Analysis: C.J.O., A.R.B. Data discussion and review: A.R.B., C.J.O., C.L., D.B., L.G., S.G., A.P.C., C.R., M.S. Writing—original draft: A.R.B., C.J.O., D.B., A.P.C., C.L., L.G., S.G., C.R. Writing—review & editing: A.R.B., C.J.O., C.L., D.B., L.G., S.G., A.P.C., C.R., M.S.

## Competing interests
The authors declare no competing interests.
