## [Peer Review File · Communications Biology]

Reviewers' comments:

Reviewer #1 (Remarks to the Author):

The paper presents a global spatial analysis of carnivore habitat (range) and potential impact on individual livestock keepers if livestock were depredated. The core issues of human-wildlife conflict and coexistence are extremely relevant. While the paper's framing and goals represent a potentially novel contribution and would be of broad interest, I cannot recommend publication in the current form. Chief among concerns are that the data and analysis do not support conclusions, primarily due to the lack of observations of livestock depredation or even livestock or human presence.

I'm uncertain whether the authors can address the central shortcoming with the analysis: no livestock depredation is observed. Further, no data on livestock distribution are included. While the approach is transparent - loss is considered at the individual scale simply as if a single calf (TLU) was depredated by carnivores - this has a number of implications for how the data are analyzed and presented. What does it mean that actual depredation risk is not assessed, treating areas of true high risk/impact the same as areas of no impact? Further, risk is assessed simply as carnivore presence, with no data on human or livestock presence. Addressing the concern will require more than extreme caution in language, particularly when comparing across nations (e.g., the opening paragraph of Results; of the leading heading of Discussion: "The poorest bear the largest economic cost of human-wildlife conflict," while such conflicts are not observed or analyzed).

A second area of concern regards data incongruity. The authors must discuss their approach to assessing food security (and risk of insecurity due to livestock depredation) in diverse populations including agropastoralists and pastoralists who rarely consume or sell cattle. While the paper makes comparisons across developing, transitioning, and developed economies, little to no attention is given to cultural differences among livestock keeping groups that significantly shape interactions with carnivores. In general, the paper must engage with the diversity of pastoralist and livestock keeping livelihoods. The current Discussion makes poor pastoralists central but could be interpreted as based on very thin generalizations, particularly given the global scope. See important work by Dyson-Hudson, Galvin, Ellis, McCabe, Herrero, Thornton; see Wiederkehr, Moritz, and Liao for more recent work.

Regarding the framing of the argument and defense of the approach, there are 2 significant reference omissions. First, Di Minin et al recently published a study on risk of crop and livestock depredation across Africa using a similar but distinct spatially explicit approach (Nat Comm). It's important for the authors to engage with these methods and defend their own choices of analysis and data. Second, Salerno et al have published a number of recent multinational papers from the Kavango-Zambezi, including on food security impacts of wildlife conflicts (Current Bio, Con Bio) and disproportionate burden of living with wildlife and unequal management authority (Con Letters), mostly situated in the context of the SDGs.

Smaller but still significant points

I suggest omitting the "Economic disparities in national cow production" analysis. FAO data are based on national livestock markets and do not accurately represent differences in production systems between, for example, cow-calf operations in Montana exposed to risk of grizzly predation and Samburu pastoralists around Ewaso. As presented, the analysis masks these differences and does not really relate to carnivore data.

It is difficult to evaluate the paper in the current format, which is not that of Nature Communications Biology. Reformatting will involve more than pasting Methods to the end of the paper.

Much of the Discussion strays widely from the data and findings. I think it's important to adhere to the analysis of carnivore occurrence. Livestock depredation is not observed, and even claims regarding risk of depredation are perhaps unsupported when livestock and human presence are not observed. The sections regarding the SDGs and debt and tax restructuring stray too far from the data.

Small points

- Include spp/scientific names where relevant
- Discussion section on SDGs should engage with SDGs at the outside and make concrete links between them and the analysis. Currently, those links are less clear (eg, the first 3/4ths of the section text does not mention SDGs).
- Revisit use of cow vs cattle

Reviewer #2 (Remarks to the Author):

This is an important, global-scale paper on the economic costs of human-wildlife conflict and its disproportionate burden on the world's most impoverished people. I thank the reviewers and editors for the opportunity to review it. In my opinion, this is a well-devised analysis of global relevance and will be of great interest to many researchers in conservation, sustainable development, and scholars and practitioners engaged with issues of human-wildlife conflict and coexistence. For transparency, I am writing this review from the perspective of someone who has studied human-wildlife conflicts and their political and economic effects and consequences, but less so as an expert of spatially explicit global-scale analyses of the kind presented here. I mention this because it is possible that I may have misunderstood some aspect of the analysis or overlooked possible issues in the study design, and wish to communicate this to both the authors and editor.

With those caveats out of the way, my comments are actually quite minor because I think the authors have presented a very strong paper and analysis with some important findings that support the results of a wide swath of more empirical and case-study based literature showing that the world's most impoverished people often suffer the greatest burdens of living with large carnivores, which presents a serious challenge to their conservation.

My primary recommendation for the authors is to make clearer some of the limitations of this kind of global scale analysis and the necessary coarseness (or indirectness- e.g. night time lights) of datasets employed; and what that limits in terms of interpreting their findings. They go some way in doing this in the last section of the paper, but I think it is worth keeping in mind, for instance, that the paper is only (per my reading) assessing economic damages based on the value of an animal's meat (or per their caloric estimates, milk as well), when in many parts of the world this may not even represent the primary economic value of the animal (for instance, it could be the animal is primarily used to produce and sell dung, or milk, or as farm-labor). This more than likely means (as they describe in their findings) that their assessment is conservative, which is good, but also just speaks to some of the challenges of doing this kind of global-scale work which largely by necessity have to ignore socio-cultural differences across regions and societies that mediate the value of an animal to a pastoralist, farmer, etc.

Second, there is also a missed opportunity as I see it to bring these findings 'down to earth' so to speak in terms of how their results might look different across different geographic regions- not just based on a flattening of the world through per capita income: Do these results demand additional interpretation when we compare these costs between SE Asia, India, and say, Tanzania? I suspect the authors have little room for additional text, but if possible, it could be a welcome addition to the paper. That said, I love the figures provided here.

Third, do the authors feel the need to say anything or respond to the specific framing of the paper through the lens of human-wildlife conflict, given advances in discussions and debates around human-wildlife coexistence and how the term human-wildlife conflict can also mask how many perceived conflicts are in fact conflicts between humans over management and access to resources, etc?

Again, this was a welcome read based on what appears to be a tremendous amount of effort; I agree with the author's overall conclusions and findings based on their results, and the above comments are intended to serve the authors in improving the paper in some modest ways, its framing, and its overall impact and relevance to others.

Yours Sincerely,

Jared Margulies

Some additional minor comments and suggestions:

Line 87-88: Minor, but perhaps rephrase sentence so that it doesn't sound like livestock predation necessarily leads to retaliatory killings - this isn't always the case, of course, and varies across cultures, etc

252-253: This is surprising (re: cattle pricing same across developing and developed and transitioning economies), though I see where it is coming from based on the FAO dataset. As I'm not an expert on FAO data, I'm just curious if there is anything about the dataset itself the authors feels need mentioning in terms of potential limitations here. I.E.- for a non expert in FAO data reading this paper, is there anything we need to know about how that data is 'cooked' that may be influencing subsequent analyses?

271: Our results confirm that not only (suggested edit)

313- suggest replacing paradoxical with 'at times conflicting'

318-321: This point about tourism reducing conflict is perhaps too generalizing, which is of course a common issue in global scale analyses of these kind. There are a lot of studies that also show that ecotourism revenue so often does not actually reach the target population you are describing here- more often than not, it is elites in many regions that benefit from tourism rather than pastoralists, farmers, etc. So often this question of where money from tourism is derived is tied to issues of land tenure and land rights, which represent vastly different questions and problems across different world regions based on tenure/land ownership structures. I think this needs to be offered as a caveat or clarified.

As just one example off the top of my head that shows how marginal some of these tourism benefits can be (in the context of South India):

Karant, K. K., & DeFries, R. (2011). Nature-based tourism in Indian protected areas: New challenges for park management. *Conservation Letters*, 4(2), 137-149.

380: This comment ties into my larger point above. In many regions of the world as I am sure the authors are familiar, the economic value of livestock may be multiple and have compounding effects on livelihoods, especially if the animal is predominantly valued for both economic and nutritive value of milk, but also dung production for fertilizer- either for use or for direct sale. It is great to see the authors also mentioning issues around these gendered dynamics/burdens/costs of HWC, as well as potential psychological costs. But providing a reference as just one example of these compounding economic impacts of loss (which are also mediated by cattle type, which is perhaps something that deserves just a little mention in the paper, or how this analyses might have been altered if they authors conducted this analysis using a goat as the measure of loss rather than a cow):

Margulies, J. D., & Karant, K. K. (2018). The production of human-wildlife conflict: A political animal geography of encounter. *Geoforum*, 95, 153-164.

Reviewer #3 (Remarks to the Author):

This is a brilliant and succinct article that highlights the unequal and inequitable burden of human-carnivore conflict globally. This was a joy to read and sheds an incredibly bright spotlight on how developing nations shoulder the extraordinary brunt of carnivore-livestock conflict. Both the results

and the discussion section do a great job of emphasizing the opposing forces of UN SDGs, the developed world's desire to save charismatic large carnivores, and the potential lived financial ramifications experienced on the ground by farmers/pastoralists in developing nations. I would argue that this will be a defining narrative for future research in this realm as we begin to grapple with how conservation biology and global environmental justice wrestle with an ever-changing landscape driven by climate change.

I only have a few comments – mainly areas where there were a few typos, but that's about it. Kudos to the authors, this is an exceptional piece.

Line 239: I would suggest converting calories values to kcal, since the numbers in calories are so large. I would also suggest doing the same for calorie values in the rest of this paragraph

Lines 271-273: This sentence reads a bit awkward; try revising.

Lines 274-276: This is an important and critical point. I would suggest adding 1-2 sentences about how dire this is in the context of global climate change, and extreme perturbations to annual rainfall, that is likely to serve as a negative multiplier of the effects you've observed here.

Lines 294-297: This is brilliant, please keep this

Line 314: Missing the hyphen in "human-wildlife conflict"

Lines 349-350: Of the many reasons included in this paragraph, one that is not touched on (and could help to strengthen the argument being made even further) is the fact that some of these estimates are made on data collected from 2009 - nearly 12 years ago. Chances are the loss of a calf is higher now than it was in 2009, meaning that the economic loss for individuals is greater, despite a tepid increase in income for most of those farmers/pastoralists. Global economic disruptions due to the pandemic are also likely to have contributed to major changes in how much money each individual is able to get for each cattle, further amplifying the extreme burden faced by folks in developing regions.

Lines 347-379: I would suggest breaking this paragraph up into two different paragraphs. There is a lot of content in here, but some of it gets lost with how large the paragraph is. I would also suggest ending the paper generally with 2-4 sentences on how this work is relevant in a global climate change context. So perhaps rather than emphasizing that context in lines 274-276, you can conclude with that narrative here to bring the paper to a close.

Reviewer #1 (Remarks to the Author):

The paper presents a global spatial analysis of carnivore habitat (range) and potential impact on individual livestock keepers if livestock were depredated. The core issues of human-wildlife conflict and coexistence are extremely relevant. While the paper's framing and goals represent a potentially novel contribution and would be of broad interest, I cannot recommend publication in the current form. Chief among concerns are that the data and analysis do not support conclusions, primarily due to the lack of observations of livestock depredation or even livestock or human presence.

Thank you for your comment. It is important to clarify here that our paper did not quantify global rates of predation on livestock by large carnivores, and this was done by design. Our analysis concerns the relative consequences of a single predation event - if one occurs - in different regions and economic circumstances. This question can be addressed - as we have done here - separately from the question of how frequent or likely that occurrence is.

Observations or estimates of predation rate, or of human or animal population, whilst relevant, are therefore not actually necessary for our analysis and conclusions. In economics, we distinguish between "extensive margin" and "intensive margin". In this context it means that if a predation event happens, we have damage of X (extensive margin). We cannot say anything about the intensive margin (how often attacks happen and how many cattle are affected, and how high the probability of these events is). For clarity, we now use the term vulnerability instead of risk throughout our manuscript.

I'm uncertain whether the authors can address the central shortcoming with the analysis: no livestock depredation is observed. Further, no data on livestock distribution are included. While the approach is transparent - loss is considered at the individual scale simply as if a single calf (TLU) was depredated by carnivores - this has a number of implications for how the data are analyzed and presented. What does it mean that actual depredation risk is not assessed, treating areas of true high risk/impact the same as areas of no impact? Further, risk is assessed simply as carnivore presence, with no data on human or livestock presence. Addressing the concern will require more than extreme caution in language, particularly when comparing across nations (e.g., the opening paragraph of Results; of the leading heading of Discussion: "The poorest bear the largest economic cost of human-wildlife conflict," while such conflicts are not observed or analyzed).

As our response above we no longer use the term risk, instead we use the term vulnerability throughout the manuscript. We recognise that the term risk was misleading in our first version due to not looking at the predation frequency.

We agree that including livestock distribution is important and we thank the reviewer for this point. We now account for the presence of cattle by masking our large carnivore layers with the most comprehensive global spatial dataset on cattle distribution (Gilbert et al., 2018). We found that the overlap between the cattle distribution and our livestock layers was >95%.

A second area of concern regards data incongruity. The authors must discuss their approach to assessing food security (and risk of insecurity due to livestock depredation) in diverse populations including agropastoralists and pastoralists who rarely consume or sell cattle. While the paper makes comparisons across developing, transitioning, and developed economies, little to no attention is given to cultural differences among livestock keeping groups that significantly shape interactions with carnivores. In general, the paper must engage with the diversity of pastoralist and livestock keeping livelihoods. The current Discussion makes poor pastoralists central but could be interpreted as based on very thin generalizations, particularly given the global scope. See important work by Dyson-Hudson, Galvin, Ellis, McCabe, Herrero, Thornton; see Wiederkehr, Moritz, and Liao for more recent work.

We agree that many agropastoralists do not sell or even consume cattle products – take for instance the Banyankole in western Uganda, or Maasai across large swathes of Kenya. Both tribal groups keep cattle for cultural reasons, as a store of value, and largely subsist of the renewable blood and milk products from their cattle (rarely killing them). These tribal groups are likely to respond to cattle predation by carnivores in varying ways, even by province or district. However, what we have emphasized in this revision is that some households may be buffered against the effects of carnivore predation due to owning large aggregations of cattle or engaging in large-scale production ranches (like in Brazil's Pantanal) – these are very different from nomadic pastoralists, say in northern Kenya or Uganda. We have provided the following discussion of this statement in lines 312-332:

“How agropastoralists view and utilise cattle also varies sharply across the globe, and as Dyson-Hudson and Dyson-Hudson (1980) note in their seminal paper “there is an enormous variability in herd management strategies, in social organization, in land tenure, degree of dependence on agricultural products, interactions with outside groups, differentiations of sex and age, etc”. We agree that such variation may be present even at incredibly fine spatial scales. For example, the Bahima and Karamajong pastoralists of Uganda's Mbarara and Karamoja districts, place immense cultural and monetary value into cattle keeping. It is a central part of their identity (Barber 2009). They largely subsist off cow milk and blood, use cows as dowry or enzhugano (Oberg 1949), but rarely slaughter the cows for meat (Purseglove et al. 1939). Contrastingly, South African Zulus slaughter a high quantity of cattle for consumption in their diet (Canonici 1991). Similar contrasts can be observed in India. Roughly 80% of the human population is comprised of Hindus (UN 2022) which consider cattle as sacred and do not slaughter and consume cattle. This contrasts with the roughly 14% of Indians which are Muslim. 40% of these Muslims include cattle meat in their diet. Our analysis does not capture the nuances of these cultural or even religious valuations of cattle. We also do not differentiate the variations in production practice of cattle globally. For example, in the Brazillian Pantanal cattle ranches are large (over half are between 5-30 000 hectares in size, Walfrido et al. 2019), intensive in nature, and slaughter and export to the international market (Vale et al. 2019). This contrasts with for example Maasai group ranches in southeast Kenya which own on average between 250-650 hectares depending on their location (De Leeuw et al. 1984). Or analysis encapsulates both highly productive intensive systems, and nomadic, subsistence production types.”

Regarding the framing of the argument and defense of the approach, there are 2 significant reference omissions. First, Di Minin et al recently published a study on risk of crop and

livestock depredation across Africa using a similar but distinct spatially explicit approach (Nat Comm). It's important for the authors to engage with these methods and defend their own choices of analysis and data. Second, Salerno et al have published a number of recent multinational papers from the Kavango-Zambezi, including on food security impacts of wildlife conflicts (Current Bio, Con Bio) and disproportionate burden of living with wildlife and unequal management authority (Con Letters), mostly situated in the context of the SDGs.

We agree that the inclusion of these references will strengthen the manuscript. We now cite both of these key papers in the manuscript, in lines 114 and 216. We have also cited the majority of the requested works highlighted above. These new citations can be found in the following locations of the new version:

Di Minin, E., Slotow, R., Fink, C., Bauer, H., & Packer, C. (2021). A pan-African spatial assessment of human conflicts with lions and elephants. *Nature Communications*, 12(1), 1-10. Cited in lines 114.

Salerno, J., Stevens, F. R., Gaughan, A. E., Hilton, T., Bailey, K., Bowles, T., ... & Hartter, J. (2021). Wildlife impacts and changing climate pose compounding threats to human food security. *Current Biology*, 31(22), 5077-5085. Cited in line 216.

Dyson-Hudson, N., & Dyson-Hudson, R. (1982). The structure of East African herds and the future of East African herders. *Development and Change*, 13(2), 213-238. Cited in line 83.

Dyson-Hudson, R., & Dyson-Hudson, N. (1980). Nomadic pastoralism. *Annual Review of Anthropology*, 15-61. Cited in line 313.

Galvin, K. A. (2009). Transitions: pastoralists living with change. *Annual Review of Anthropology*, 38(1), 185-198. Cited in line 206.

Stavi, I., Roque de Pinho, J., Paschalidou, A. K., Adamo, S. B., Galvin, K., de Sherbinin, A., ... & van der Geest, K. (2021). Food security among dryland pastoralists and agropastoralists: The climate, land-use change, and population dynamics nexus. *The Anthropocene Review*, 20530196211007512. Cited in line 207.

Herrero, M., Addison, J., Bedelian, C., Carabine, E., Havlík, P., Henderson, B., ... & Thornton, P. K. (2016). Climate change and pastoralism: impacts, consequences and adaptation. *Rev Sci Tech*, 35(2), 417-433. Cited in lines 84 and 205.

Smaller but still significant points

I suggest omitting the "Economic disparities in national cow production" analysis. FAO data are based on national livestock markets and do not accurately represent differences in production systems between, for example, cow-calf operations in Montana exposed to risk of grizzly predation and Samburu pastoralists around Ewaso. As presented, the analysis masks these differences and does not really relate to carnivore data.

We agree that the production systems are different across countries and acknowledge that a 10000-hectare ranch in Montana may experience different vulnerability to predation when compared to a nomadic Samburu family in Kenya. Our justification for including this analysis was to illustrate that developing economies face an added economic shock, not only from the loss of the cattle calf (financial and caloric loss) but also are at a disadvantage because they produce less cattle meat to begin with. To address the reviewer's concern we now add a caveat in the discussion regarding the FAO data. Lines 375-384.

It is difficult to evaluate the paper in the current format, which is not that of Nature Communications Biology. Reformatting will involve more than pasting Methods to the end of the paper.

Much of the Discussion strays widely from the data and findings. I think it's important to adhere to the analysis of carnivore occurrence. Livestock depredation is not observed, and even claims regarding risk of depredation are perhaps unsupported when livestock and human presence are not observed. The sections regarding the SDGs and debt and tax restructuring stray too far from the data.

We thank the reviewer for their helpful comments. We have now removed the term risk from our analysis which we agree was misleading (this has been replaced with the correct term of vulnerability). We now also incorporate the cattle distribution layer in our analysis as requested. We believe that our analysis is now more reflective of reality, especially in how large carnivores and cattle overlap.

Regarding the point on debt and tax restructuring, we have now removed this entire section and heading "*Implementing development solutions that draw value from costly carnivores*". Our inclusion of reference to the SDG's is meant to highlight the potential conflicts that arise from communities living alongside predators deemed valuable nationally or globally but have significant economic impacts locally. Our reference to the SDG's in the context of our analysis is made simply to illuminate the above point of these being in conflict with one another.

We have included a call for conservationists and policymakers to use our results as a further motivation to seek novel financing mechanisms to bridge the economic disparities and valuations of large carnivores globally (vs locally). We specifically highlight the recent South African rhino bond and Swedish payment for population performance programme aimed at conserving the wolverine. This is found in lines 271-286.

Small points

-Include spp/scientific names where relevant

Included throughout in parentheses for all species.

-Discussion section on SDGs should engage with SDGs at the outside and make concrete links between them and the analysis. Currently, those links are less clear (eg, the first 3/4ths of the section text does not mention SDGs).

Please see our comments above on the SDG's.

-Revisit use of cow vs cattle

We have now replaced 'cow' with 'cattle' throughout.

Reviewer #2 (Remarks to the Author):

This is an important, global-scale paper on the economic costs of human-wildlife conflict and its disproportionate burden on the world's most impoverished people. I thank the reviewers and editors for the opportunity to review it. In my opinion, this is a well-devised analysis of global relevance and will be of great interest to many researchers in conservation, sustainable development, and scholars and practitioners engaged with issues of human-wildlife conflict and coexistence. For transparency, I am writing this review from the perspective of someone who has studied human-wildlife conflicts and their political and economic effects and consequences, but less so as an expert of spatially explicit global-scale analyses of the kind presented here. I mention this because it is possible that I may have misunderstood some aspect of the analysis or overlooked possible issues in the study design, and wish to communicate this to both the authors and editor.

With those caveats out of the way, my comments are actually quite minor because I think the authors have presented a very strong paper and analysis with some important findings that support the results of a wide swath of more empirical and case-study based literature showing that the world's most impoverished people often suffer the greatest burdens of living with large carnivores, which presents a serious challenge to their conservation.

We are grateful for your encouraging review of our paper. We have made the requested revisions below.

My primary recommendation for the authors is to make clearer some of the limitations of this kind of global scale analysis and the necessary coarseness (or indirectness- e.g. night time lights) of datasets employed; and what that limits in terms of interpreting their findings. They go some way in doing this in the last section of the paper, but I think it is worth keeping in mind, for instance, that the paper is only (per my reading) assessing economic damages based on the value of an animal's meat (or per their caloric estimates, milk as well), when in many parts of the world this may not even represent the primary economic value of the animal (for instance, it could be the animal is primarily used to produce and sell dung, or milk, or as farm-labor). This more than likely means (as they describe in their findings) that their assessment is conservative, which is good, but also just speaks to some of the challenges of doing this kind of global-scale work which largely by

necessity have to ignore socio-cultural differences across regions and societies that mediate the value of an animal to a pastoralist, farmer, etc.

We are grateful for the comment and suggestion. This point mirrors the observations of reviewer 1 in that our original version did not articulate in enough detail that both production intensity of cattle and how cattle are viewed culturally and even religiously will differ markedly across space. We have discussed both points in detail in our new version, paying attention to examples from Uganda, India, Brazil, Kenya, and South Africa. We have added a new paragraph to discuss these points, which is located in the “Burden estimates are conservative and only illustrate the spatial economic vulnerability of HWC”. It is quoted above in response to reviewer 1’s comments (this document, page 4).

Second, there is also a missed opportunity as I see it to bring these findings 'down to earth' so to speak in terms of how their results might look different across different geographic regions- not just based on a flattening of the world through per capita income: Do these results demand additional interpretation when we compare these costs between SE Asia, India, and say, Tanzania? I suspect the authors have little room for additional text, but if possible, it could be a welcome addition to the paper. That said, I love the figures provided here.

We agree with the reviewer that it can be helpful to bring such global analyses “down to earth”. We believe that our supplementary tables highlight many of these individual country-level disparities, moreover our central Figure 1 also emphasizes these stark contrasts.

Third, do the authors feel the need to say anything or respond to the specific framing of the paper through the lens of human-wildlife conflict, given advances in discussions and debates around human-wildlife coexistence and how the term human-wildlife conflict can also mask how many perceived conflicts are in fact conflicts between humans over management and access to resources, etc?

We thank the reviewer for this point. We feel that this has been largely addressed in the “Global birthright or local burden: the conflicting nature of the UN Sustainable Development Goals” section. Here we discuss varying valuation systems surrounding conflict-causing predators that are valued largely for tourism and ecosystem services by international audiences and national governments, but these valuations are not necessarily shared by people living alongside them. This is again reemphasized in lines 233-286 with the recent work of Jacobsen et al 2022. in Hwange, Zimbabwe.

Again, this was a welcome read based on what appears to be a tremendous amount of effort; I agree with the author's overall conclusions and findings based on their results, and the above comments are intended to serve the authors in improving the paper in some modest ways, its framing, and its overall impact and relevance to others.

Yours Sincerely,

Jared Margulies

Some additional minor comments and suggestions:

Line 87-88: Minor, but perhaps rephrase sentence so that it doesn't sound like livestock predation necessarily leads to retaliatory killings - this isn't always the case, of course, and varies across cultures, etc

We agree and have tempered this statement, highlighting several case studies of cultural and religious factors playing into the protection of large carnivores. The specific adjustment is located in lines 89-93:

“In areas where cultural and religious valuations of carnivores are absent (eg. Li et al. 2014, Bhatia et al. 2016, Gebresenbet et al. 2017) conflict between agricultural communities and carnivores often escalates carnivore extinction risk due to the retaliatory killings following livestock predation (Cardillo et al. 2004; Hazzah et al. 2009; Plaza et al. 2019; Mateo-Tomás and López-Bao 2020).”

252-253: This is surprising (re: cattle pricing same across developing and developed and transitioning economies), though I see where it is coming from based on the FAO dataset. As I'm not an expert on FAO data, I'm just curious if there is anything about the dataset itself the authors feels need mentioning in terms of potential limitations here. I.E.- for a non expert in FAO data reading this paper, is there anything we need to know about how that data is 'cooked' that may be influencing subsequent analyses?

We have specified what this data is and how it was collected in the “Estimating Cattle Prices” section in lines 375-384:

“We obtained data on national cattle prices per kilogram (Item Code 945, FAO) from the Food and Agriculture Organization of the United Nations (FAO 2021). This dataset reports on cattle prices as collected at the point of initial sale (prices paid at the farm-gate).”

271: Our results confirm that not only (suggested edit)

Now changed in line 201.

313- suggest replacing paradoxical with 'at times conflicting'

Changed in the heading and also in text to conflicting.

318-321: This point about tourism reducing conflict is perhaps too generalizing, which is of course a common issue in global scale analyses of these kind. There are a lot of studies that also show that ecotourism revenue so often does not actually reach the target population you are describing here- more often than not, it is elites in many regions that benefit from tourism rather than pastoralists, farmers, etc. So often this question of where money from tourism is derived is tied to issues of land tenure and land rights, which represent vastly different questions and problems across different world regions based on tenure/land ownership structures. I think this needs to be offered as a caveat or clarified.

As just one example off the top of my head that shows how marginal some of these tourism benefits can be (in the context of South India):

Karanth, K. K., & DeFries, R. (2011). Nature-based tourism in Indian protected areas: New challenges for park management. *Conservation Letters*, 4(2), 137-149.

We have revised this statement to the following (lines 263-269), including the suggested citation:

“In cases where human communities living alongside protected areas and damage-causing wildlife realize the monetary value from these species (e.g., through ecotourism revenues, conservancies, and population performance payments (Zabel and Holm-Müller 2008)), and this revenue is not subjected to elite capture or is marginal in nature (Platteau 2004, Karanth and DeFries 2011), pressures on protected areas and wildlife wane due to increases in human development indices and escape from poverty traps (eg. Ament et al. 2019, Geldmann et al. 2019, Naidoo et al. 2019).”

380: This comment ties into my larger point above. In many regions of the world as I am sure the authors are familiar, the economic value of livestock may be multiple and have compounding effects on livelihoods, especially if the animal is predominantly valued for both economic and nutritive value of milk, but also dung production for fertilizer- either for use or for direct sale. It is great to see the authors also mentioning issues around these gendered dynamics/burdens/costs of HWC, as well as potential psychological costs. But providing a reference as just one example of these compounding economic impacts of loss (which are also mediated by cattle type, which is perhaps something that deserves just a little mention in the paper, or how this analyses might have been altered if they authors conducted this analysis using a goat as the measure of loss rather than a cow):

Margulies, J. D., & Karanth, K. K. (2018). The production of human-wildlife conflict: A political animal geography of encounter. *Geoforum*, 95, 153-164.

We have now included a statement in lines 223-231:

“Our analysis also illustrates two sources of additional and lost opportunity cost, that of useable calories from meat, and those garnered from milk and associated products. We estimate the immediate useable calories to be somewhere between roughly half a year and 1.5 years of calories depending on human age, while we estimate an additional ~300 kg of milk and 12 kg of meat/year from cow yield. These kinds of costs ignore additional valuations of other cattle products such as dung used for fertiliser (Madhusudan 2005, Margulies and Karanth 2018), the hidden mental costs associated with the loss of a sacred animal in some regions (eg. Simoons et al. 1981), and critical thresholds of cattle herd size (Maystadt and Ecker 2014).”

Reviewer #3 (Remarks to the Author):

This is a brilliant and succinct article that highlights the unequal and inequitable burden of human-carnivore conflict globally. This was a joy to read and sheds an incredibly bright spotlight on how developing nations shoulder the extraordinary brunt of carnivore-livestock conflict. Both the results and the discussion section do a great job of emphasizing the opposing forces of UN SDGs, the developed world's desire to save charismatic large carnivores, and the potential lived financial ramifications experienced on the ground by farmers/pastoralists in developing nations. I would argue that this will be a defining narrative for future research in this realm as we begin to grapple with how conservation biology and global environmental justice wrestle with an ever-changing landscape driven by climate change.

We are grateful for the support of the reviewer for our paper, we have made changes requested below.

I only have a few comments – mainly areas where there were a few typos, but that's about it. Kudos to the authors, this is an exceptional piece.

Line 239: I would suggest converting calories values to kcal, since the numbers in calories are so large. I would also suggest doing the same for calorie values in the rest of this paragraph

These changes are made throughout the manuscript – kilocalories throughout.

Lines 271-273: This sentence reads a bit awkward; try revising.

This has been revised to the following in lines 201-203:

“Our results confirm that not only is this true in absolute per capita income loss, but also due to pastoralists in developing economies producing less meat per animal and having a lower meat intake per capita.”

Lines 274-276: This is an important and critical point. I would suggest adding 1-2 sentences about how dire this is in the context of global climate change, and extreme perturbations to annual rainfall, that is likely to serve as a negative multiplier of the effects you've observed here.

We have rewritten large parts of this paragraph to the following in lines 203-223:

“This is notable because in many areas of the developing world pastoralists are already under immense pressure from localized rainfall patterns and drought (Maystadt and Ecker 2014) and this is causing livestock units per capita to decline (Stavi et al. 2021). This in turn has a knock on effect on herd size which when below a critical threshold leads pastoralism to no longer be viable (Lybbert et al. 2004, Maystadt and Ecker 2014). Estimates vary but typically if a pastoralist owns ≤ 2 adult cattle (4.5 tropical livestock units equivalent to 1125 kg of livestock biomass per capita) they are unable to maintain a nomadic pastoralist lifestyle (Maystadt and Ecker 2014). Any source of livestock loss at this point is too severe for economic recovery (Lybbert et al. 2004) and will potentially erase the wealth of an entire

pastoralist family. These income losses have a knock-on effect on food security, and this may disproportionately affect women, who forgo meals to keep the household fed (Ogra 2008, Botreau and Cohen 2019). This mirrors crop raiding scenarios like those highlighted by Salerno et al. (2021) who recently showed this pressure in the Kavango-Zambezi Transfrontier Conservation Area. These authors showed that elephant-associated crop raiding exacerbated the likelihood of food insecurity due to an already reduced rainfall environment. We anticipate that such knock on effects we discuss here are likely experienced by many if not most of the households that would lose $\geq 25\%$ of per capita income in our analysis. Moreover, further knock-on effects are likely, such as migration, subsequent social, and political instability, and impacts to childhood cognitive skills and education due to loss of calories (Prado and Dewey 2014).”

Lines 294-297: This is brilliant, please keep this

We have retained this statement.

Line 314: Missing the hyphen in “human-wildlife conflict”

Now added.

Lines 349-350: Of the many reasons included in this paragraph, one that is not touched on (and could help to strengthen the argument being made even further) is the fact that some of these estimates are made on data collected from 2009 - nearly 12 years ago. Chances are the loss of a calf is higher now than it was in 2009, meaning that the economic loss for individuals is greater, despite a tepid increase in income for most of those farmers/pastoralists. Global economic disruptions due to the pandemic are also likely to have contributed to major changes in how much money each individual is able to get for each cattle, further amplifying the extreme burden faced by folks in developing regions.

We have adjusted the reasoning behind conservative estimates to the following (lines 292-311):

We feel however, that our analysis is extremely conservative for several reasons including: 1) we only show the potential economic ramifications of a single predation event. Carnivore depredation often manifests itself in specific areas due to habitat or ecological variables (Miller 2015, Gastineau et al. 2019), households often experience predation events multiple times per year (Dickman et al. 2011), and sometimes carnivores engage in surplus killing (Kruuk 1972, Khorozyan et al. 2017, Lucherini et al. 2018). Our analysis does not calculate actual depredation rates, it only illustrates highly vulnerable (and buffered) economic areas globally that would suffer under a predation event, 2) our valuation of a depredated calf is also likely low, because we adjusted the measure of FAO market cow value to the slaughter weight of a single ~6-month-old calf (250 kg), equating to roughly 33% of the economic value of an adult cow, 3) our measure of a single calf as a proportion of per capita GDP is well below the threshold of total income generated from many cattle production systems in Africa and Asia (for instance across much of sub-Saharan Africa livestock typically contributes between half and all of household income in rural settings, with cattle contributing 70-90%; Otte and Chilonda 2002). We acknowledge that a lack of regional cow

price data means that our analysis of cow market price is relatively coarse. Prices in livestock change with the breed of cattle, seasonality (and drought) and with prevailing macro and micro economic conditions (e.g., Ocaido et al. 2009), and 4) our data originate from 2009 and the acceleration of human-induced climate change, the COVID-19 pandemic, and increased rates of political strife may have impacted the severity of our analysis even further.

Lines 347-379: I would suggest breaking this paragraph up into two different paragraphs. There is a lot of content in here, but some of it gets lost with how large the paragraph is. I would also suggest ending the paper generally with 2-4 sentences on how this work is relevant in a global climate change context. So perhaps rather than emphasizing that context in lines 274-276, you can conclude with that narrative here to bring the paper to a close.

We have included the suggestions of reviewer 1 and also reviewer 2 in completely expanding this section to also include the fact that our analysis does not consider the nuances of 1) how cattle are viewed from cultural and religious perspectives, and 2) how widely production practices differ between regions. For this reason we have separated our paragraph into three distinct sections: 1) key reasons for why our analysis is conservative (including your point on COVID, increasing climate induced effects on cattle production and political strife), 2) differing perspectives on cattle production and how cattle are viewed from cultural and religious perspectives, and 3) we discuss hidden costs, lost opportunity costs such as human fear, risk of sickness, and political issues surrounding claiming compensation. The new section reads as follows in lines 290-346:

Our analysis of the potential economic burden arising from losing livestock to carnivores was only possible due to recent developments in global socioeconomic (Lessmann and Seidel 2017) and carnivore range spatial data (Brooks et al. 2019). We feel however, that our analysis is extremely conservative for several reasons including: 1) we only show the potential economic ramifications of a single predation event. Carnivore depredation often manifests itself in specific areas due to habitat or ecological variables (Miller 2015, Gastineau et al. 2019), households often experience predation events multiple times per year (Dickman et al. 2011), and sometimes carnivores engage in surplus killing (Kruuk 1972, Khorozyan et al. 2017, Lucherini et al. 2018). Our analysis does not calculate actual depredation rates, it only illustrates highly vulnerable (and buffered) economic areas globally that would suffer under a predation event, 2) our valuation of a depredated calf is also likely low, because we adjusted the measure of FAO market cow value to the slaughter weight of a single ~6-month-old calf (250 kg), equating to roughly 33% of the economic value of an adult cow, 3) our measure of a single calf as a proportion of per capita GDP is well below the threshold of total income generated from many cattle production systems in Africa and Asia (for instance across much of sub-Saharan Africa livestock typically contributes between half and all of household income in rural settings, with cattle contributing 70-90%; Otte and Chilonda 2002). We acknowledge that a lack of regional cow price data means that our analysis of cow market price is relatively coarse. Prices in livestock change with the breed of cattle, seasonality (and drought) and with prevailing macro and micro economic conditions (e.g., Ocaido et al. 2009), and 4) our data originate from 2009 and the acceleration of human-induced climate change, the COVID-19 pandemic, and

increased rates of political strife may have impacted the severity of our analysis even further.

How agropastoralists view and utilise cattle also varies sharply across the globe, and even at incredibly fine spatial scales. For example, the Bahima and Karamajong pastoralists of Uganda's Mbarara and Karamoja districts, place immense cultural and monetary value into cattle keeping. It is a central part of their identity (Barber 2009). They largely subsist off cow milk and blood, use cows as dowry or *enzhugano* (Oberger 1949), but rarely slaughter the cows for meat (Purseglowe et al. 1939). Contrastingly, South African Zulus slaughter a high quantity of cattle for consumption in their diet (Canonici 1991). Similar contrasts can be observed in India. Roughly 80% of the human population is comprised of Hindus (UN 2022) which consider cattle as sacred and do not slaughter and consume cattle. This contrasts with the roughly 40% of the Muslim population which include meat in their diet. Our analysis does not capture the nuances of above cultural and even religious valuations of cattle. We also do not differentiate the variations in production practice of cattle globally. For example, in the Brazilian Pantanal cattle ranches are large (over half are between 5-30 000 hectares in size, Walfrido et al. 2019), intensive in nature, and slaughter and export to the international market (Vale et al. 2019). This contrasts with for example Maasai group ranches in southeast Kenya which own on average between 250-650 hectares depending on their location (De Leeuw et al. 1984). Our analysis encapsulates both highly productive intensive systems, and nomadic, subsistence production types.

Finally, our results also only touch on the many potential opportunity costs (Figure 3) stemming from conflict. There are a multitude of hidden costs (Barua et al. 2013) and perceived conflicts that can be associated with such a loss including increased workload to make up for financial losses (Ogra 2008), physical displacement of households (e.g., Choudhury 2004), the physical and disease exposure risks of guarding livestock at night to prevent further losses (Barua et al. 2013), and transaction costs of pursuing compensation payments and the failure to obtain fair livestock value (Sherman and Dixon 1990, Brackowski et al. 2020c). A recent example from Zimbabwe shows that the presence of an African lion equates to negative USD\$180 per person per year due to fear, and a lack of trust in compensation authorities responsible in mitigating conflict events (Jacobsen et al. 2022). There may also be a host of psychological effects stemming from such conflict including fear of attack by carnivores, hesitancy to move in the dark, grief over lost livestock, and even PTSD from livestock loss, not to mention the immense cost of human life itself (Gulati et al. 2021).

Reviewers' comments:

Reviewer #1 (Remarks to the Author):

I appreciate the authors' attention to my comments and the associated revisions. Their clarification on the risk v vulnerability distinction makes sense and I think now makes for a more justifiable argument. The further discussion and references are also appreciated and I think address concerns and strengthen the paper. There are a few remaining issues that I think need to be addressed.

The explanation the authors gave in their response regarding distinguishing between extension and intensive margin makes sense, however I still feel that the tone of the paper must shift following a much more explicit recognition that carnivore predation of livestock risk, or burden, or impacts are not being examined. As I said above, I appreciate the shift to 'vulnerability,' but a shift in the tone must follow to support a more honest and transparent framing and discussion. For example, L115-117 presents this work as a global comparison of economic burden of large carnivores... even if it's qualified by a later statement of potential cost. At the outset of Results, L134 states that data show that the world's poorest 'bear the highest cost of living with carnivores.' Again I appreciate that the following sentence clarifies it's the potential cost, but because of the features of the analysis (which I'm fine with) the authors must be extremely cautious and precise with statements of findings and their interpretations. Another example is in referring to 'conflict burden areas' (L154) - I know the meaning is ambiguous between conflict experience and conflict vulnerability (the latter as independent from exposure) but the point is that all ambiguity must be removed. I stress that these are not the only cases where statements must be corrected.

Relatedly, I think the title should be changed to reflect the analysis more directly.

The new discussion paragraph beginning L271 introduces a very good point. But I'd encourage the authors to consider additional models to reduce vulnerability, specifically non-economic or compensatory models such as (potential) conflict reduction. This could be a brief few sentences pointing to a few examples of ongoing efforts.

Smaller points

L38: It's subtle, but 'disparities *from* conflict are not show,' only disparities in vulnerability to conflict if it were to occur.

L43: I'd replace 'pastoralists' with cattle keepers throughout, since many cattle keepers do not engage in pastoralism, mobile or otherwise. There are other places where pastoralist is used but doesn't match the associated production system(s).

Throughout, there's some inconsistency in decimal places presented, along with a few "," and "." errors. A quick copy edit should clear them up.

L186: "cattle" should likely be "cow" if it's a per animal evaluate

Reviewer #2 (Remarks to the Author):

I want to thank the editor, other reviewers, and authors for the opportunity revisit this paper and learn from their revisions and the other reviewer reports. Overall I think the authors have done a great job responding to both mine and other reviewer criticisms. I am pleased with their revisions and think they've done a thorough and clear job responding to them. Again, as a caveat, I cannot speak to how Reviewer #1 or the editor assess how the authors have responded to their concerns, as that is less my area of expertise. I am satisfied with these revisions in relation to the specific issues and questions I raised.

One very small point- the revised text includes the word 'seminal'. I would replace this, as it is very gendered language.

I think this is a valuable paper and think the authors have done a good job qualifying some of the limits to this kind of coarse scale analysis while also demonstrating their value.

Reviewer #1 (Remarks to the Author):

I appreciate the authors' attention to my comments and the associated revisions. Their clarification on the risk v vulnerability distinction makes sense and I think now makes for a more justifiable argument. The further discussion and references are also appreciated and I think address concerns and strengthen the paper. There are a few remaining issues that I think need to be addressed.

We appreciate the reviewer's constructive feedback on our latest revision. We have made renewed effort on this second revision to address each suggestion.

The explanation the authors gave in their response regarding distinguishing between extension and intensive margin makes sense, however I still feel that the tone of the paper must shift following a much more explicit recognition that carnivore predation of livestock risk, or burden, or impacts are not being examined. As I said above, I appreciate the shift to 'vulnerability,' but a shift in the tone must follow to support a more honest and transparent framing and discussion. For example, L115-117 presents this work as a global comparison of economic burden of large carnivores... even if it's qualified by a later statement of potential cost.

We have tempered the above section by stating “potential economic burden” – this is related to our change of wording to “economic vulnerability”. The change is located in lines 144-147: “no work to date has comparatively explored the potential economic burden of large carnivores globally. Here we present the first spatially explicit analysis of the potential economic burden arising from human-wildlife conflict at the global scale illustrating the financial and social costs of losing livestock on human communities.”

We have also taken care to temper much of the general tone throughout the manuscript by explicitly using the terms “potential burden” (eg. In line 43 of the abstract, in line 144 of the main text, in line 167 and 175 of the results section, and line 320 of the discussion), using the word “may” to suggest when a conflict event occurs (eg. line 169 of the Results), and also acknowledging that our paper is an examination of the potential economic ramifications of conflict globally (lines 367-368).

At the outset of Results, L134 states that data show that the world's poorest 'bear the highest cost of living with carnivores.' Again I appreciate that the following sentence clarifies it's the potential cost, but because of the features of the analysis (which I'm fine with) the authors must be extremely cautious and precise with statements of findings and their interpretations. Another example is in referring to 'conflict burden areas' (L154) - I know the meaning is ambiguous between conflict experience and conflict vulnerability (the latter as independent from exposure) but the point is that all ambiguity must be removed. I stress that these are not the only cases where statements must be corrected.

We have attempted to remove ambiguity by defining exactly what our analysis has done – this may be found in lines 141-147, please note the clear reference to disparities in economic vulnerability and potential economic burden:

“Using an analysis based on subnational 1st level administrative regions our data show that the world's poorest people may bear the highest cost of living with large carnivores (Figure 1). This is based on the disparities in economic vulnerability to carnivore predation on cattle. People living in developing countries will on average experience an eightfold higher potential economic burden (\bar{x} =32%, range = 0.02-201% of per capita income lost in a large carnivore predation event) than those living in developed economies (\bar{x} =4% income lost, range=1-9%, Supplementary Table 2).”

Please also see our response above giving the exact locations of where and how we have tempered our language to better represent the economic vulnerability theme of our paper.

Relatedly, I think the title should be changed to reflect the analysis more directly.

The new discussion paragraph beginning L271 introduces a very good point. But I'd encourage the authors to consider additional models to reduce vulnerability, specifically non-economic or compensatory models such as (potential) conflict reduction. This could be a brief few sentences pointing to a few examples of ongoing efforts.

We have now added in lines 334-337 the following on non-economic models:

“Indeed, human-wildlife conflict has also been shown to reduce using non-economic conflict reduction models. These include inter alia the introduction of human and canine guardians for livestock (e.g., McManus et al. 2015), and the building of protective enclosures and bomas to shield livestock against predators (Lichtenfeld et al. 2015).

Smaller points

L38: It's subtle, but 'disparities *from* conflict are not show,' only disparities in vulnerability to conflict if it were to occur.

We have made this correction as suggested. We now also temper your point on burden by replacing it with “potential burden” throughout the manuscript.

L43: I'd replace 'pastoralists' with cattle keepers throughout, since many cattle keepers do not engage in pastoralism, mobile or otherwise. There are other places where pastoralist is used but doesn't match the associated production system(s).

We have replaced pastoralists with cattle keepers throughout the manuscript.

Throughout, there's some inconsistency in decimal places presented, along with a few "," and "." errors. A quick copy edit should clear them up.

We are grateful for the detection of this error. We have checked the entire manuscript for consistency and made the requested changes.

L186: "cattle" should likely be "cow" if it's a per animal evaluate.

In the last revision reviewers requested that cow be changed to cattle throughout the manuscript for consistency. FAO data also does not discern the sex of the animal – we therefore maintain the term “cattle” throughout.

Reviewer #2 (Remarks to the Author):

I want to thank the editor, other reviewers, and authors for the opportunity revisit this paper and learn from their revisions and the other reviewer reports. Overall I think the authors have done a great job responding to both mine and other reviewer criticisms. I am pleased with their revisions and think they've done a thorough and clear job responding to them. Again, as a caveat, I cannot speak to how Reviewer #1 or the editor assess how the authors have responded to their concerns, as that is less my area of expertise. I am satisfied with these revisions in relation to the specific issues and questions I raised.

We are grateful to the reviewer for their reassessment of our new version and their kind words. Their suggestions have strengthened the paper greatly.

One very small point- the revised text includes the word 'seminal'. I would replace this, as it is very gendered language.

We have replaced the word seminal with the word “key” in line 397.

I think this is a valuable paper and think the authors have done a good job qualifying some of the limits to this kind of coarse scale analysis while also demonstrating their value.

We are grateful for the reviewer comment.